# Deciding Technosignature Search Strategies: Multi-Criteria Fuzzy Logic to Find Extraterrestrial Intelligence

**Juan Miguel Sánchez-Lozano** [1,*] , **Eloy Peña-Asensio** [2,3] **and Hector Socas-Navarro** [4,5]

1   Centro Universitario de la Defensa, Universidad Politécnica de Cartagena, C/ Coronel López Peña S/N, Base Aérea de San Javier, Santiago de La Ribera, 30720 Murcia, Murcia, Spain
2   Departament de Química, Universitat Autònoma de Barcelona, 08193 Bellaterra, Catalonia, Spain; eloy.peas@gmail.com
3   Institut de Ciències de l'Espai (ICE, CSIC), Campus UAB, C/ de Can Magrans S/N, 08193 Cerdanyola del Vallès, Catalonia, Spain
4   Instituto de Astrofísica de Canarias, Avda Vía Láctea S/N, 38205 La Laguna, Tenerife, Spain; hsocas@iac.es
5   Departamento de Astrofísica, Universidad de La Laguna, 38205 La Laguna, Tenerife, Spain
*   Correspondence: juanmi.sanchez@cud.upct.es

**Abstract:** This study presents the implementation of Multi-Criteria Decision-Making (MCDM) methodologies, particularly the fuzzy technique for order of preference by similarity to ideal solution (TOPSIS), in prioritizing technosignatures (TSs) for the search for extraterrestrial intelligence (SETI). By incorporating expert opinions and weighted criteria based on the established Axes of Merit, our analysis offers insights into the relative importance of various TSs. Notably, radio and optical communications are emphasized, in contrast to dark side illumination and starshades in transit. We introduce a new axis, Scale Sensitivity, designed to assess the variability of TS metrics. A sensitivity analysis confirms the robustness of our approach. Our findings, especially the highlighted significance of artifacts orbiting Earth, the Moon, or the Sun, indicate a need to broaden evaluative criteria within SETI research. This suggests an enhancement of the Axes of Merit, with a focus on addressing the plausibility of TSs. As the quest to resolve the profound question of our solitude in the cosmos continues, SETI efforts would benefit from exploring innovative prioritization methodologies that effectively quantify TS search strategies.

**Keywords:** technosignatures; search for extraterrestrial intelligence (SETI); technique for order preference by similarity to ideal solution (TOPSIS); analytic hierarchy process (AHP); linguistic labels; alternatives; criteria

## 1. Introduction

The investigation into physical manifestations of extraterrestrial technological activity constitutes a specialized subfield between astrobiology and astrophysics, aimed at addressing one of the most profound existential questions: Are we alone in the cosmos? Termed technosignatures [1], hereinafter TSs, their identification has the potential to illuminate not merely the existence of extraterrestrial life, but also the evolutionary trajectories of life forms beyond Earth.

The quest for TSs falls under the umbrella of the search for ixtraterrestrial intelligence (SETI). Since the seminal work by Cocconi and Morrison in the late 1950s, which posited the detectability of interstellar communications [2], the scope of TS searches has expanded well beyond traditional radio frequencies [3]. This expansion has been further catalyzed by the intersection with biosignatures, thereby augmenting the roster of potential TSs. According to a recent report by NASA Science Mission Directorate [4], TSs may even be more prevalent than habitable exoplanets. Encouraging this line of inquiry, Socas-Navarro et al. [5] put forth a framework for prospective missions, advocating for both public and private space agencies to formulate targeted strategies for TS detection. Among

the indicators under investigation are industrial gases in atmospheric spectra, thermal emissions from megastructures, Earth-orbiting artifacts, and specific optical and radio frequency signals.

However, formulating and evaluating an effective search strategy in SETI is a complex endeavor. This complexity arises from the multitude of TSs involved and the challenge of establishing a unified search strategy. Indeed, each researcher in the field of SETI may advocate for unique methodologies, thereby exacerbating the difficulty in achieving community-wide consensus [6].

In this context, the 2018 NASA TS Workshop at the Lunar and Planetary Institute (LPI) in Houston, USA, laid the foundational groundwork for achieving such a consensus [6]. The workshop introduced a new framework based on a set of nine attributes, termed Axes of Merit, designed to address questions from both practical and scientific standpoints. These questions include the level of technological development required for the search, associated costs, potential auxiliary benefits, and the ease of TS detection, among others (we will detail them later).

According to Sheikh [7], the Axes of Merit cannot serve as quantitative measures for evaluating TSs, given that even the weights assigned to each axis are influenced by subjective factors. Consequently, the pertinent question emerges: How can one prioritize TSs according to their respective Axes of Merit values, especially when these axes may have associated variable weights of importance? The resolution to this conundrum can be found in established engineering project methodologies. Specifically, decision theory-based methodologies, when integrated with techniques such as fuzzy logic, can manage this uncertainty and address such decision-making challenges [8]. These methodologies are known as fuzzy adaptations of Multi-Criteria Decision-Making (MCDM) approaches.

While the integration of MCDM methodologies with fuzzy logic is not novel in the fields of astronomy and astrophysics, as evidenced by various studies [9–13], its application to the realm of SETI is unprecedented. This approach enables the prioritization of existing TSs based on the nine Axes of Merit, as proposed by Wright [6]. Consequently, it addresses the challenge posed by Sheikh [7] by allowing for the weighting of each Axis of Merit and facilitating qualitative evaluations of the current TSs.

To accomplish this, fuzzy adaptations of two well-established MCDM methods are employed. The Analytic Hierarchy Process (AHP), proposed by Saaty [14], is used to determine the weights or importance coefficients of the criteria (Axes of Merit in this context), while the technique for order of preference by similarity to ideal solution (TOPSIS), developed by Hwang and Yoon [15], serves to prioritize the various alternatives (both categories of TSs and individual TSs). To this end, a questionnaire based on both methodologies has been completed by an advisory group of SETI experts.

The structure of this paper is as follows. Section 2 elucidates the concept of fuzzy sets (Section 2.1) and outlines the two MCDM methodologies—AHP (Section 2.2) and TOPSIS (Section 2.3)—applied to address this decision problem. Section 3 provides a brief overview of the case study, detailing the TSs under evaluation and the criteria for prioritizing alternatives (Axes of Merit), along with the data acquisition methods. Sections 4 and 5 present the results yielded by this approach, and Section 6 summarizes the key conclusions drawn from our study and proposes avenues for future research.

## 2. Methodology

In this study, we select a combination of fuzzy versions of MCDM methodologies (AHP and TOPSIS) due to their complementary strengths. AHP is utilized for its ability to decompose any decision-making problem into a structured hierarchy encompassing objectives, criteria, and alternatives, thereby simplifying the complexity of the problem. Its use of pairwise comparisons for criteria also adds an intuitive element to the evaluation process. Concurrently, the TOPSIS method is employed for its logical framework, which facilitates the representation of both the criteria involved in the decision-making process and their respective significance coefficients through straightforward mathematical procedures.

The suitability and justification of the proposed methodology is detailed in the respective sections of this article.

### 2.1. Fuzzy Sets

On countless occasions, human beings must select from several options which one is the best. It is also common that, when selecting between these options, there are a set of criteria to consider. MCDM methodologies have been postulated as ideal techniques to address this type of decision problem in disciplines as diverse as energy [16,17], medicine [18], logistics [19], or even astronomy [20,21]. However, it is also prevalent to have to address decision problems in which some criteria are difficult to quantify. That is where fuzzy logic emerges as a powerful tool, as it can manage the lack of certainty about the true values of the data and the parameters, that is, it can handle its uncertainty.

Since their creation in the 1960s [22], fuzzy sets have been applied in numerous branches of science, even combined with MCDM approaches [8]. Its use is especially appropriate in situations where there is no strict threshold to define whether an object or individual is classified in one category or another. For example, in terms of height, we could consider that an individual belongs to the "very tall" category if he/she measures more than 2.00 m, while others could assign individuals whose height is 1.85 m in the same category. The starting point of any fuzzy set is the domain of a membership function on the unit interval $[0, 1]$. The level of membership is measured through a set of numbers in that interval so that, the closer the value of an object is to 1, the higher its level of membership in a certain category. Likewise, the closer it is to 0, the more unlikely it is that such an object belongs to said category.

From a mathematical point of view, let $A$ be a fuzzy set in the universe of discourse $U$ which contains a collection of points (or objects) denoted by $X$. Thus, a membership function, $f_A : U \to [0, 1]$, can be described as a mathematical rule that assigns each element $x \in U$ to the degree of membership of $x$ to $A$, $f_A(x) \in [0, 1]$. In the evaluation processes of alternatives with multiple criteria in which the values of the criteria are not precise, due to their subjective nature, such valuations can be performed through linguistic variables associated with fuzzy sets based on this type of membership function.

Throughout the years, numerous membership functions of different types have been developed to reflect the preferences of decision-makers, starting with the classic triangular, Gaussian, sigmoidal, or trapezoidal membership functions [23], to the most recent extensions such as spherical fuzzy functions [24], without leaving aside the intuitionistic [25], Pythagorean [26], picture [27], or neutrosophic [28] fuzzy functions. In fact, the preferences of decision-makers have been frequently analyzed in the literature; current examples are the models for managing incomplete information and consensus with hesitant fuzzy linguistic preference relations in group decision-making problems [29], new approaches based on multi-granular hesitant fuzzy linguistic term sets [30], or even extended logarithmic least squares methods to derive a priority weight vector from a fuzzy preference relation with self-confidence [31].

Although these functions, and even new extensions, are being developed today, the advantage of using some over others has not yet been demonstrated; even recent studies have shown that new extensions of fuzzy MCDM versions generate greater dependence on the judgments provided by decision-makers [32]. Also, starting from the premise that it is unnecessary to increase the complexity of the calculation process, fuzzy sets based on triangular membership functions are applied in this study. This type of function not only simplifies the calculation process due to its easy handling but also fits with the way of representing the qualitative criteria (described in Section 3.1). Moreover, the application of the fuzzy versions, based on triangular fuzzy numbers, of the MCDM methodologies proposed in this study has already been used to solve study cases as diverse as the selection of onshore wind farms [33], evaluation of near-Earth asteroid deflection techniques [10], or the assessment of international military high-performance aircraft [34].

A triangular fuzzy function presents three parameters $(a, b, c)$, which correspond to the three vertices of a triangle, and is specified by the following expression:

$$triangle(x; a, b, c) = \begin{cases} 0, & if \quad x \leq a \\ \frac{(x-a)}{(b-a)} & if \quad a < x \leq b \\ \frac{(c-x)}{(c-b)} & if \quad b < x \leq c \\ 0, & if \quad c > x \end{cases} \tag{1}$$

In this work, we use triangular membership functions to evaluate, via linguistic labels based on an importance scale, the alternatives of the decision problem (TSs) for each one of the criteria that influence the evaluation process. A representation of these labels is shown in Figure 1.

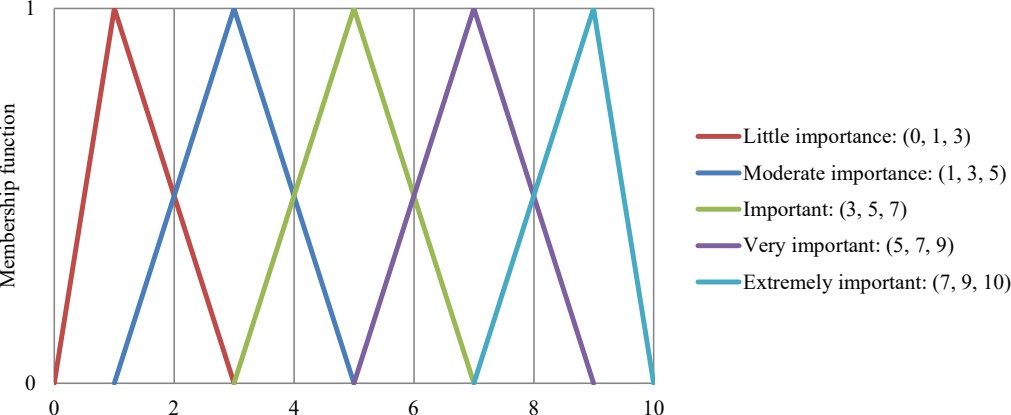

**Figure 1.** Example of triangular membership function defined by linguistic labels based on importance scale.

### 2.2. The Fuzzy Analytic Hierarchy Process (AHP)

The AHP is an MCDM methodology developed by Thomas L. Saaty in the 1980s [14]. This MCDM methodology allows any decision problem to be represented through a hierarchical structure with three or more levels: the objective or goal to be achieved constitutes the upper level of the hierarchy, the criteria and sub-criteria that influence the evaluation problem are in the following levels, and finally, the alternatives to be evaluated are represented at the bottom of the hierarchy.

Through a process of paired comparison between elements located at the same level of the hierarchy, AHP generates the weights or importance coefficients of the criteria and sub-criteria. This process is based on a comparison scale called the Saaty scale, whose fuzzy version [35] is shown in Table 1.

**Table 1.** Fuzzy scale of valuation in the pair-wise comparison process.

| Labels | Verbal Judgments of Preferences | Triangular Fuzzy Scale |
|---|---|---|
| EqI | $C_i$ and $C_j$ are Equally Important | (1, 1, 1) |
| Sl + I | $C_i$ is Slightly More Important than $C_j$ | (2, 3, 4) |
| +I | $C_i$ is More Important than $C_j$ | (4, 5, 6) |
| St + I | $C_i$ is Strongly More Important than $C_j$ | (6, 7, 8) |
| Ex + I | $C_i$ is Extremely More Important than $C_j$ | (8, 9, 9) |

Such a comparison scale is used by decision-makers or experts to generate a comparison matrix $H$ of dimensions $n \times n$ where the pairs of compared criteria $(C_i, C_j)$ are represented. Likewise, the $H_{12}$ value corresponds to the relative importance of $C_1$ to $C_2$, $H_{12} \approx (w_1/w_2)$. The four properties, which the matrix $H$ satisfies, can be defined as follows:

1. $h_{ij} \approx \left({}^{w_i}/_{w_j}\right) \quad i, j = 1, 2, \ldots, n.$
2. $h_{ii} = 1 \quad i = 1, 2, \ldots, n.$
3. If $h_{ij} = \alpha, \alpha \neq 0$, then $h_{ji} = \left({}^{1}/_{\alpha}\right) \quad i, j = 1, 2, \ldots, n.$
4. If $C_i$ is more important than $C_j$, then $h_{ij} \approx \left({}^{w_i}/_{w_j}\right) > 1.$

According to the previous rules, the matrix $H$ is reciprocal and positive, with $1's$ in its main diagonal. Furthermore, in determining the weights of the criteria whose nature is qualitative, the normalized geometric mean based on triangular fuzzy numbers can be applied:

$$w_i = \frac{\prod_{j=1}^{n}(a_{ij}, b_{ij}, c_{ij})^{\left({}^{1}/_{n}\right)}}{\sum_{i=1}^{m}\prod_{j=1}^{n}(a_{ij}, b_{ij}, c_{ij})^{\left({}^{1}/_{n}\right)}}, \tag{2}$$

where $(a_{ij}, b_{ij}, c_{ij})$ is a triangular fuzzy number.

This expression, represented by triangular fuzzy numbers, allows us to approximate the calculation of the eigenvector of the matrix $H$, directly obtaining the weight vector. In this study, the fuzzy version of the AHP methodology is applied to obtain the weights of the criteria that influence the evaluation of each alternative. Such criteria correspond to the nine Axes of Merit for TS searches [6] plus one extra axis, which are detailed in Section 3.1.

### 2.3. Technique for Order of Preference by Similarity to Ideal Solution (TOPSIS)

TOPSIS (for the technique for order of preference by similarity to ideal solution) is, after the AHP methodology, the second most widespread and applied MCDM methodology [8]. This technique, which was developed by Hwang and Yoon [15], allows generating a ranking of alternatives through the relative proximity of each alternative to an ideal solution. Its underlying idea consists of choosing the best alternative as the one that simultaneously keeps the greatest distance from the negative ideal solution (based on cost criteria) and the lowest distance from the positive ideal solution (based on benefit criteria). As such, the relative proximity or optimal solution provided by TOPSIS becomes a compromise answer concerning the preferences of the decision-maker.

A fuzzy version of the TOPSIS approach becomes especially appropriate when dealing with decision problems involving criteria of a qualitative nature (as is the case of the decision problem of this study). Throughout this paper, the weights of the criteria (which quantify their relative importance) as well as their ratings are expressed in terms of linguistic variables through triangular fuzzy numbers (see Table 1 and Figure 1).

Next, the computational steps of the TOPSIS algorithm, which is shown in Figure 2, are described.

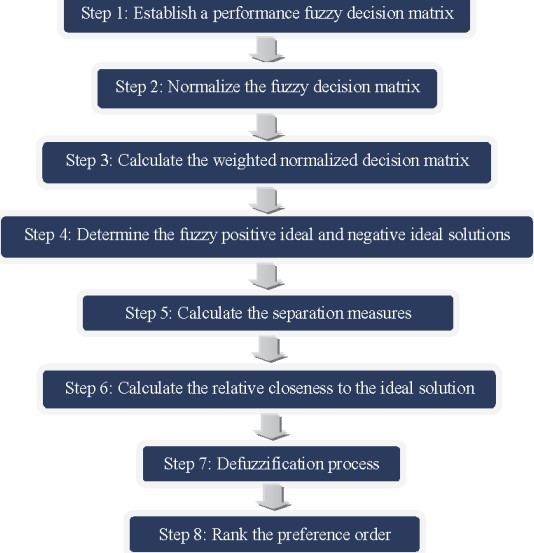

**Figure 2.** Scheme of all the stages in the fuzzy TOPSIS approach.

- Step 1. Establish a performance fuzzy decision matrix (Table 2). It is composed of rows and columns so that the rows constitute the alternatives $A_i$ $(i = 1, 2, \ldots, m)$ to be evaluated and the columns show the valuations of each of the criteria $C_j$ $(j = 1, 2, \ldots, n)$ that influence the assessment process for each one of the alternatives. In this study, such valuations are obtained via linguistic labels through triangular fuzzy numbers (see Figure 1). For simplicity of representation, the stages of the TOPSIS algorithm are shown with real values or crisp numbers. The only difference concerning the fuzzy version lies in applying the arithmetic operations of triangular fuzzy numbers, detailed in [36,37].

**Table 2.** The decision matrix.

| MCDM | $C_1$ | $C_2$ | $C_3$ | $C_n$ |
|------|-------|-------|-------|-------|
| $A_1$ | $x_{11}$ | $x_{12}$ | $\ldots$ | $x_{1n}$ |
| $A_2$ | $x_{21}$ | $x_{22}$ | $\ldots$ | $x_{2n}$ |
| $\ldots$ | $\ldots$ | $\ldots$ | $x_{ij}$ | $\ldots$ |
| $A_m$ | $x_{m1}$ | $x_{m2}$ | $\ldots$ | $x_{mn}$ |

- Step 2. Normalize the fuzzy decision matrix. The decision matrix defined in the previous step is normalized through the following expression:

$$n_{ij} = \frac{x_{ij}}{\sqrt{\sum_{i=1}^{m} x_{ij}^2}}, \tag{3}$$

where $n_{ij}$ corresponds to an element of the normalized decision matrix $N$.

- Step 3. Calculate the weighted normalized decision matrix. Once the global weight vector of the criteria is obtained (via the fuzzy version of the AHP methodology in our decision problem), the normalized decision matrix is multiplied by such a vector as follows:

$$v_{ij} = w_j \otimes n_{ij}, j = 1, \ldots, n, i = 1, \ldots, m, \tag{4}$$

where $w_j's$ satisfy $\sum_{j=1}^{n} w_j = 1$

- Step 4. Determine the fuzzy positive ideal $A^+$ and negative ideal $A^-$ solutions. The characteristics of the criteria determine both solutions since, in the case of a benefit criterion, the $A^+$ solution will correspond to the maximum valuation of the alternatives for such a criterion. Likewise, if the criterion is cost-based, the $A^+$ value will be obtained through the minimum valuation. An analogous but inverse case occurs to determine the $A^-$ solution. Mathematically, such concepts ($A^+$ and $A^-$) are defined through the following:

$$A^+ = \{ v_1^+, \ldots, v_n^+ \} = \begin{cases} max_i\{ v_{ij}, j \in J \}, i = 1, 2, \ldots, m & \text{if criterion is to maximize} \\ min_i\{ v_{ij}, j \in J' \}, i = 1, 2, \ldots, m & \text{if criterion is to minimize} \end{cases} \tag{5}$$

$$A^- = \{ v_1^-, \ldots, v_n^- \} = \begin{cases} min_i\{ v_{ij}, j \in J \}, i = 1, 2, \ldots, m & \text{if criterion is to maximize} \\ max_i\{ v_{ij}, j \in J' \}, i = 1, 2, \ldots, m & \text{if criterion is to minimize} \end{cases} \tag{6}$$

where the parameter $J'$ is associated with cost criteria, while the parameter $J$ corresponds to benefit criteria.

- Step 5. Calculate the separation measures of each alternative. The separation distances of each alternative from $A^+$ (namely, $d_i^+$) and $A^-$ (namely, $d_i^-$) are determined by the n-dimensional Euclidean distance method:

$$d_i^+ = \sqrt{\sum_{j=1}^{n} (v_{ij} - v_j^+)^2} \quad for \ \ i = 1, 2, \ldots, m \tag{7}$$

$$d_i^- = \sqrt{\sum_{j=1}^{n}(v_{ij} - v_j^-)^2} \quad for \quad i = 1, 2, \ldots, m \tag{8}$$

- Step 6. Calculate the relative closeness to the ideal solution. Hence, a closeness coefficient for each alternative can be calculated as follows:

$$CC_i = \frac{d_i^-}{d_i^+ + d_i^-} \quad for\ all \quad i = 1, 2, \ldots, m, \tag{9}$$

- Step 7. Defuzzification process. Since each closeness coefficient is a triangular fuzzy number in the fuzzy version of the TOPSIS methodology, that coefficient must be transformed into a crisp one. To deal with such a defuzzification process, we shall apply the following expression:

$$I_{\frac{1}{3},\frac{1}{2}}(CC_i) = \frac{1}{3}\frac{a_1 + 4b_1 + c_1}{2}. \tag{10}$$

This defuzzification index allows us to rank fuzzy numbers by modality and optimistic of the decision-maker attitude. Its suitability has been demonstrated thanks to its application in previous studies [33,34,36]. More detailed information about such an index can be seen at [38].

- Step 8. Rank the preference order. It becomes clear that a given alternative $A_i$ stands closer to the $A^+$ and farther from the $A^-$ (and hence, its ranking is higher) as $CC_i$ becomes closer to 1. Therefore, a ranking of alternatives in descending order is generated according to $I_{\frac{1}{3},\frac{1}{2}}(CC_i)$ values.

The fuzzy version of the TOPSIS technique is applied in this work to evaluate all the alternatives involved in the decision problem (TSs) and generate a ranking.

## 3. Prioritization Problem: Selecting TS Search Strategies

In the quest to unravel the potential existence of extraterrestrial intelligence, the strategic selection of TS search strategies stands as a cornerstone of our scientific endeavors. This section is dedicated to systematically evaluating various TS alternatives, scrutinized through a well-defined set of criteria. Our approach bifurcates into two comprehensive subsections, each addressing a different aspect of the TS search.

First, we outline the Axes of Merit that underpin our assessment. These encompass nine specific criteria, augmented by an additional axis named "Scale Sensitivity", which we define in Section 3.1. This set of criteria forms a robust framework, enabling us to evaluate the viability and effectiveness of different TS search strategies comprehensively.

Following this, we delve into the alternatives themselves. Our analysis adopts a dual-layered approach; initially, we evaluate groups of TS alternatives collectively, gaining insight into their aggregate potential and implications in the broader quest for extraterrestrial intelligence. Subsequently, we transition to a detailed, individualized examination of each TS alternative. This methodical approach ensures a thorough and nuanced understanding, capturing both the collective synergy and the distinctive features of each alternative in our quest to detect signs of advanced civilizations beyond our planet.

### 3.1. Definition of the Criteria

We summarize the criteria that form the bedrock of our assessment methodology, constituting the nine Axes of Merit plus an additional axis we coin as "Scale Sensitivity", collectively referred to as the criteria ($C_i$). These criteria are devised to provide a comprehensive framework for evaluating the potential and efficacy of various TS search strategies.

- $C_1$—Observing Capability: This criterion evaluates the current technological capabilities of astronomy in detecting specific TSs. It encompasses the sophistication of

observational equipment, the extent of astronomical knowledge, and the ability to distinguish between natural cosmic phenomena and potential TSs.

- $C_2$—Cost: A multifaceted criterion that encompasses the financial investment required, the allocation of telescope time, computational resources, and other opportunity costs associated with the search.
- $C_3$—Ancillary Benefits: Every search for alien civilizations should be structured in a manner that yields valuable scientific data, regardless of the success in discovering extraterrestrial life. This criterion appraises the capacity of a TS search strategy to contribute to broader scientific knowledge.
- $C_4$—Detectability: This involves assessing the strength of the TS signal relative to background cosmic noise. A key factor here is the signal's clarity and distinctness, which significantly impacts the likelihood of accurate detection.
- $C_5$—Duration: The temporal aspect of a TS's detectability is evaluated here. This criterion measures the length of time for which a TS remains observable.
- $C_6$—Ambiguity: This criterion assesses the likelihood of a TS being erroneously interpreted as a natural phenomenon unrelated to extraterrestrial life. Minimizing ambiguity is crucial for the validity of any potential discovery.
- $C_7$—Extrapolation: Here, the focus is on our capability to comprehend, interpret, and potentially utilize the underlying technology of a detected TS, should it be within the realms of our current or near-future technological understanding.
- $C_8$—Inevitability: This criterion evaluates the likelihood that a given technology, used by an advanced civilization, would naturally produce a detectable TS. It reflects the probability that certain technologies are universal and would manifest detectable signatures.
- $C_9$—Information: This axis gauges the richness of the information carried by a detected TS, including the quantity and diversity of data. It reflects the potential scientific value embedded within the signal.
- $C_{10}$—Scale Sensitivity: Introduced as an additional axis, this criterion quantifies the variability in the assessment of the other merit axes as a function of the TS's size or intensity, within a plausible expected range.

To elucidate the concept of "Scale Sensitivity", consider two examples. Imagine the scale range of a Dyson sphere, extending from one built around the smallest or coldest observed star to one encompassing the largest or hottest star. This criterion seeks to capture the degree to which evaluations along the merit axes vary between these two polar examples. Another example involves examining the scale range of an atmosphere infused with industrial gases, extending from a concentration comparable to Earth's atmosphere at the onset of the industrial era to an atmosphere heavily laden with gases indicative of a runaway greenhouse effect.

The pivotal question here is how sensitive the merit axis evaluations are to such extremes. Do the assessments fluctuate significantly with changes in the scale or intensity of the TS? The "Scale Sensitivity" criterion is designed to encapsulate and measure this variability, offering a dynamic lens through which the potential impact and discernibility of varying TSs can be evaluated.

### 3.2. Description of the Alternatives

We now examine the alternatives to be evaluated using the previously established criteria. Our analysis is twofold. We first assess groups of TS alternatives collectively, gaining insights into their overall potential in the search for extraterrestrial intelligence. We then shift our focus to individual TSs, offering a detailed evaluation of each. This dual approach ensures a thorough understanding of both the collective capabilities and unique characteristics of each alternative.

First, the evaluation by categorizing potential TSs into distinct groups of alternatives ($AG_i$) is performed. These categories are outlined as follows:

- $AG_1$—Radio and Optical Communications: This category encompasses the search for technologically generated electromagnetic signals. The focus is on identifying emissions that exhibit temporal or frequency characteristics atypical of natural astrophysical sources. The approach capitalizes on the premise that advanced civilizations may utilize specific electromagnetic wavelengths for communication, which would manifest as anomalies when contrasted against the cosmic electromagnetic background.
- $AG_2$—Waste Heat: This segment explores the detection of alien megastructures and technological artifacts located beyond the confines of our solar system. The primary indicator for such structures is the thermal emission, or 'waste heat,' that they generate, potentially discernible in the infrared spectrum. These emissions would stand out from the background galactic noise, offering tangible evidence of technologically advanced civilizations.
- $AG_3$—Solar System Artefacts: This category is focused on the discovery of non-human, technologically crafted objects, substances, patterns, or processes that exist within the solar system. The premise here is that traces or remnants of extraterrestrial visitations or activities might be found closer to home, embedded within the fabric of our solar system. The search in this domain involves scrutinizing celestial bodies, including moons, asteroids, and planets, for anomalies that do not align with known natural processes or formations.

Following the collective assessment of TS groups, we shift to a more detailed evaluation of individual TS alternatives. This examination is pivotal in discerning the specific attributes and potential of each TS in the context of the criteria. The list of individual TSs (alternatives; $A_i$) we analyze includes:

- $A_1$—Industrial Gases in Atmospheric Spectra: This TS involves the detection of non-natural gases, such as chlorofluorocarbons, in the atmospheric spectra of exoplanets. The presence of these gases could indicate industrial activities, suggesting a technologically advanced civilization.
- $A_2$—Dark Side Illumination: This category refers to the observation of artificial light sources on the dark side of an exoplanet or celestial body. Such illumination could be indicative of city lights or other forms of artificial lighting, pointing to the existence of intelligent life.
- $A_3$—Starshades in Transit: The focus here is on identifying artificial structures that transit a star, leading to dimming patterns inconsistent with natural celestial bodies. Such anomalies could hint at large-scale space constructions, possibly for energy collection or habitation.
- $A_4$—Clarke Exobelt in Transit: This TS involves the detection of a dense belt of satellites or debris around an exoplanet, akin to the concept of a 'Clarke Belt' in Earth orbit. Such a belt could signify a high level of technological development in satellite deployment and space activities.
- $A_5$—Laser Pulses: The capture of high-intensity, narrow-bandwidth light signals, potentially indicative of directed energy systems or interstellar communication efforts.
- $A_6$—Heat from Megastructures: The observation of an infrared excess emanating from a star or galaxy, suggesting widespread energy utilization. Such a scenario could point to megastructures like a Dyson sphere, indicating an advanced civilization harnessing a star's energy.
- $A_7$—Radio Signals: This involves the reception of narrow-bandwidth radio signals exhibiting patterns or characteristics unlikely to be of natural origin. These signals could be indicative of communication or other technological activities.
- $A_8$—Artifacts Orbiting Earth, Moon, or the Sun: The discovery of artificial objects in stable orbits around Earth, the Moon, or traversing interstellar space. This category also includes the identification of non-natural structures or objects on the Moon or other celestial bodies, potentially left behind by advanced civilizations.



Each of these TSs offers a unique window into the potential presence of extraterrestrial intelligence. Their assessments, juxtaposed against the nine Axes of Merit plus the Scale Sensitivity criterion, provide an in-depth and multifaceted analysis. This comprehensive approach ensures a balanced and thorough exploration of the possibilities and limitations inherent in the search for signs of advanced civilizations beyond our own.

### 3.3. Obtaining the Weights of the Criteria

To carry out the process of evaluating alternatives, it is first necessary to determine if the criteria that influence such a process have the same importance or if, on the contrary, they have different weights. A questionnaire based on the fuzzy version of the AHP methodology can answer this question, providing the criteria weights.

In this decision-making process, the criteria encompass nine Axes of Merit, along with an additional axis. To guide this process, an advisory group composed of ten recognized international experts who research TSs, SETI, and related topics, will provide the information necessary to solve the proposed prioritization problem. Although the ten decision-makers who decided to participate constitute an adequate number to undertake studies of this nature, there is no specific number of experts that must be gathered. In this type of process, where the decision problem is so specific, the expertise and knowledge of each of the experts involved is much more important than having the participation of a large number of experts. They began their collaboration by addressing the first two questions of the questionnaire, which are the following:

- *Q1: Mark the relevance order that you consider appropriate for each Axis of Merit ($C_i$) plus the criterion Scale Sensitivity to prioritize TSs. Note that multiple criteria may be chosen to be equally important.*

This question allows us to sort the criteria in descending order according to each expert's importance for each criterion. In this work, the advisory group differs slightly on the importance of the mentioned criteria; their preferences are provided in Table 3.

**Table 3.** Order of importance of criteria for each expert.

| Expert | Reported Order of Importance |
|---|---|
| $E_1$ | $C_1 > C_2 = C_4 = C_6 > C_{10} = C_8 = C_5 > C_3 > C_9 = C_7$ |
| $E_2$ | $C_1 > C_4 = C_8 > C_2 > C_5 > C_6 > C_7 > C_9 > C_{10} > C_3$ |
| $E_3$ | $C_4 > C_5 > C_1 > C_8 > C_2 > C_{10} > C_7 > C_3 > C_6 > C_9$ |
| $E_4$ | $C_4 > C_5 > C_8 > C_1 > C_2 > C_9 > C_3 > C_6 > C_{10} > C_7$ |
| $E_5$ | $C_5 > C_8 > C_6 > C_4 > C_1 > C_9 > C_3 > C_7 > C_{10} > C_2$ |
| $E_6$ | $C_1 > C_4 > C_5 > C_{10} > C_9 > C_8 > C_6 > C_3 > C_2 > C_7$ |
| $E_7$ | $C_4 > C_8 > C_6 > C_2 > C_3 > C_9 > C_{10} > C_1 > C_5 > C_7$ |
| $E_8$ | $C_1 = C_4 = C_6 = C_9 > C_2 = C_5 = C_7 = C_{10} > C_3 = C_8$ |
| $E_9$ | $C_4 > C_1 > C_8 > C_7 > C_5 > C_6 = C_{10} > C_2 > C_9 > C_3$ |
| $E_{10}$ | $C_9 > C_5 > C_3 = C_4 > C_6 > C_7 = C_8 > C_1 = C_2 = C_{10}$ |

Once the order of importance for each one of the members of the advisory group is provided, the next question must be posed:

- *Q2: Compare the criterion ($C_i$) you have ranked in the first position with those you have designated for the second position and subsequent orders. For this comparison, utilize the following specified linguistic labels, (EqI), (Sl + I), (+I), (St + I), (Ex + I), which correspond to the scale of the valuation in the pair-wise comparison process* (Table 1).

This question offers the possibility of being more precise when distinguishing the judgments of preferences among the previously ordered criteria. Each of the experts must perform this comparison process individually. To illustrate the process of obtaining the weights of the criteria, the responses of Expert 1 ($E_1$), and the subsequent steps, are provided as an example. The individual responses of $E_1$ for this second question are shown in Table 4. The meaning is as follows. Criterion $C_1$ is slightly more important than $C_2$, $C_4$,

and $C_6$, more important than $C_{10}$, $C_8$, and $C_5$, strongly more important than $C_3$, and finally, extremely more important than $C_9$ and $C_7$.

**Table 4.** Valuation given by Expert 1 ($E_1$) sorted by their order of preference.

| $E_1$ | $C_1$ | $C_2$ | $C_4$ | $C_6$ | $C_{10}$ | $C_8$ | $C_5$ | $C_3$ | $C_9$ | $C_7$ |
|---|---|---|---|---|---|---|---|---|---|---|
| $C_1$ | $EqI$ | $Sl+I$ | $Sl+I$ | $Sl+I$ | $+I$ | $+I$ | $+I$ | $St+I$ | $Ex+I$ | $Ex+I$ |

According to Table 1, Expert 1's preferences can be translated into triangular fuzzy numbers, generating Table 5.

**Table 5.** $E_1$'s valuations represented by triangular fuzzy numbers.

| $E_1$ | $C_1$ | $C_2$ | $C_4$ | $C_6$ | $C_{10}$ | $C_8$ | $C_5$ | $C_3$ | $C_9$ | $C_7$ |
|---|---|---|---|---|---|---|---|---|---|---|
| $C_1$ | $(1,1,1)$ | $(2,3,4)$ | $(2,3,4)$ | $(2,3,4)$ | $(4,5,6)$ | $(4,5,6)$ | $(4,5,6)$ | $(6,7,8)$ | $(8,9,9)$ | $(8,9,9)$ |

By performing the calculations of Equation (2) with fuzzy numbers, whose operations are detailed in [37,39], the weights of the criteria for Expert 1 are generated. Matrices shown in Table 6 provide such weights which are represented via triangular fuzzy numbers. The central or modal value of such fuzzy numbers defines the weight for each criterion.

**Table 6.** Criteria weight (represented through triangular fuzzy numbers) for expert $E_1$.

| Criteria | $C_1$ Paired Comparisons | $C_1$ Triangular Fuzzy Numbers |
|---|---|---|
| $C_1$ | $(1, 1, 1)$ | $(0.193, 0.337, 0.547)$ |
| $C_2$ | $(1/4, 1/3, 1/2)$ | $(0.052, 0.112, 0.255)$ |
| $C_3$ | $(1/4, 1/3, 1/2)$ | $(0.052, 0.112, 0.255)$ |
| $C_4$ | $(1/4, 1/3, 1/2)$ | $(0.052, 0.112, 0.255)$ |
| $C_5$ | $(1/6, 1/5, 1/4)$ | $(0.034, 0.067, 0.131)$ |
| $C_6$ | $(1/6, 1/5, 1/4)$ | $(0.034, 0.067, 0.131)$ |
| $C_7$ | $(1/6, 1/5, 1/4)$ | $(0.034, 0.067, 0.131)$ |
| $C_8$ | $(1/8, 1/7, 1/6)$ | $(0.025, 0.048, 0.089)$ |
| $C_9$ | $(1/9, 1/9, 1/8)$ | $(0.022, 0.037, 0.068)$ |
| $C_{10}$ | $(1/9, 1/9, 1/8)$ | $(0.022, 0.037, 0.068)$ |

The same process of paired comparison, previously ordering the criteria according to their importance, is carried out by each of the experts that make up our advisory group. The weights of the criteria of each one of the experts that make up the advisory group, via triangular fuzzy numbers, are shown in Table A1 of Appendix A.

Considering that all the experts in the advisory group have equal relevance in determining the weights, a homogeneous aggregation process through the arithmetic mean provides the vector of weights of the criteria that influence this decision problem (see Table 7 and Figure 3).

Analyzing the results shown in Table 7 and Figure 3, it is observed that after homogeneous aggregation, the most important criterion is $C_4$—Detectability, followed very closely by criterion $C_1$—Observational capability. The following criteria in order of importance turn out to be $C_5$—Duration and $C_9$—Information. A penultimate group is made up of criteria $C_8$—Inevitability and $C_6$—Ambiguity, and finally, the least important criteria according to the group of experts are criteria $C_2$—Cost, $C_3$—Ancillary benefits, $C_{10}$—Scale sensitivity, and $C_7$—Extrapolation. Next, such a vector of weights must be taken into consideration in the process of prioritizing alternatives, that is, when implementing the TOPSIS MCDM algorithm.

**Table 7.** Weights of criteria through experts' homogeneous aggregation.

| Criteria | Weights (Fuzzy Numbers) | Weights (%) |
|---|---|---|
| $C_1$— Observing Capability | (0.124, 0.188, 0.281) | 18.82 |
| $C_2$—Cost | (0.036, 0.058, 0.100) | 5.83 |
| $C_3$—Ancillary Benefits | (0.034, 0.053, 0.085) | 5.26 |
| $C_4$—Detectability | (0.149, 0.219, 0.327) | 21.88 |
| $C_5$—Duration | (0.077, 0.112, 0.163) | 11.24 |
| $C_6$—Ambiguity | (0.052, 0.079, 0.123) | 7.85 |
| $C_7$—Extrapolation | (0.032, 0.049, 0.077) | 4.93 |
| $C_8$—Inevitability | (0.052, 0.082, 0.132) | 8.19 |
| $C_9$—Information | (0.078, 0.108, 0.152) | 10.82 |
| $C_{10}$—Scale Sensitivity | (0.033, 0.052, 0.084) | 5.18 |

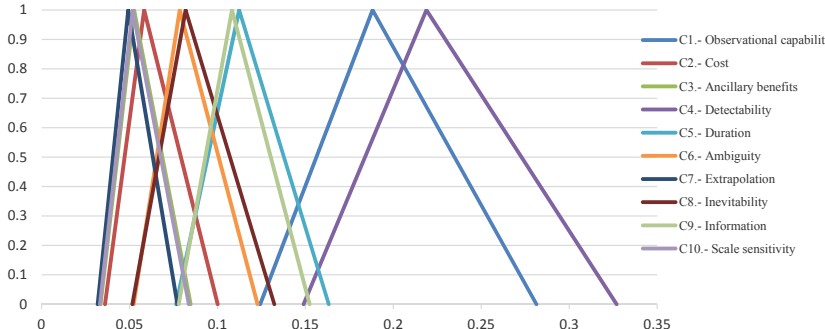

**Figure 3.** Graphical representation of the weights of the criteria by triangular fuzzy numbers.

*3.4. Prioritization of the Alternatives*

The next stage, once the weights of the criteria are obtained, corresponds to the prioritization of the alternatives. In this case study, two evaluation processes are addressed: on the one hand, the assessment of groups of alternatives (three TS categories), and on the other, the evaluation of each one of the alternatives individually (eight TSs). Both are detailed in Section 3.2.

Due to the complexity of obtaining real values of each of the criteria for each alternative or group of alternatives, the evaluations of such criteria are performed through linguistic labels associated with triangular fuzzy numbers. For this, two linguistic labels are defined. $L_1$ is used to assess the Duration Axis of Merit (criterion $C_5$), while the $L_2$ label allows the rest of the criteria to be assessed (see Table 8). In this way, our advisory group intervenes again to qualitatively assess each of the alternatives based on the set of criteria.

**Table 8.** The two linguistic labels, their gaps, and their associated triangular fuzzy numbers to assess all the alternatives by each criterion.

| Linguistic Label $L_1$ | Linguistic Label $L_2$ | Fuzzy Numbers |
|---|---|---|
| Very low (VL) | Very brief (VB) | (0, 1, 3) |
| Low (L) | Brief (B) | (1, 3, 5) |
| Medium (M) | Medium (M) | (3, 5, 7) |
| High (H) | High (H) | (5, 7, 9) |
| Very high (VH) | Very high (VH) | (7, 9, 10) |

The creation of two decision matrices that contain the qualitative valuations for each alternative and criterion, provided by the group of experts, constitutes the starting point of the fuzzy version of the TOPSIS multi-criteria methodology. Their representation through linguistic labels is shown in Tables A2 and A3 of Appendix A. Therefore, once decision matrices are defined, the fuzzy TOPSIS can be applied to obtain the relative closeness to the ideal solution of each alternative, and consequently, to generate a prioritization of alternatives.

Considering that all experts have equal relevance in the process of prioritizing alternatives, and transforming their qualitative assessments into triangular fuzzy numbers (see Table 8), two decision matrices of alternatives and criteria are generated (Tables 9 and 10); such matrices constitute step 1 of the TOPSIS algorithm.

**Table 9.** Decision matrix of alternatives and criteria through triangular fuzzy numbers (three TS categories). Note: The ranking of the categories of alternatives for each criterion is provided below the fuzzy number.

| Alternatives | Criteria | | | | |
|---|---|---|---|---|---|
| | $C_1$ | $C_2$ | $C_3$ | $C_4$ | $C_5$ |
| $AG_1$—Radio and optical communications | (5.8, 7.8, 9.1) 1st | (1.5, 3.2, 5.2) 1st | (3.5, 5.4, 7.3) 3rd | (3.7, 5.6, 7.3) 2nd | (5.4, 7.4, 8.8) 2nd |
| $AG_2$—Waste heat | (4.6, 6.6, 8.4) 2nd | (3.2, 5.2, 7.2) 2nd | (4.0, 6.0, 7.9) 2nd | (3.8, 5.8, 7.7) 3rd | (5.8, 7.8, 9.3) 1st |
| $AG_3$—Solar System artefacts | (3.5, 5.4, 7.3) 3rd | (4.6, 6.6, 8.4) 3rd | (5.2, 7.2, 8.8) 1st | (2.6, 4.4, 6.3) 1st | (5.0, 7.0, 8.6) 3rd |
| | $C_6$ | $C_7$ | $C_8$ | $C_9$ | $C_{10}$ |
| $AG_1$—Radio and optical communications | (2.0, 3.6, 5.6) 1st | (2.0, 3.6, 5.5) 1st | (3.5, 5.4, 7.1) 2nd | (5.2, 7.2, 8.8) 2nd | (3.7, 5.6, 7.4) 3rd |
| $AG_2$—Waste heat | (4.4, 6.4, 8.1) 3rd | (3.4, 5.2, 7.0) 2nd | (5.2, 7.2, 8.8) 1st | (2.6, 4.4, 6.3) 3rd | (5.4, 7.4, 9.0) 1st |
| $AG_3$—Solar System artefacts | (2.4, 4.0, 5.9) 2nd | (5.4, 7.4, 9.0) 3rd | (1.3, 3.0, 5.0) 3rd | (6.0, 8.0, 9.3) 1st | (3.9, 5.8, 7.5) 2nd |

**Table 10.** Decision matrix of alternatives and criteria through triangular fuzzy numbers (TSs individually). Note: The ranking of the alternatives for each criterion is provided below the fuzzy number.

| Alternatives | Criteria | | | | |
|---|---|---|---|---|---|
| | $C_1$ | $C_2$ | $C_3$ | $C_4$ | $C_5$ |
| $A_1$—Industrial gases on atmospheric spectra | (3.2, 5.0, 6.9) 4th | (4.1, 6.0, 7.8) 6th | (4.4, 6.4, 8.2) 2nd | (3.1, 5.0, 7.0) 5th | (4.8, 6.8, 8.5) 5th |
| $A_2$—Dark side illumination | (1.2, 2.8, 4.8) 8th | (4.7, 6.6, 8.2) 8th | (2.6, 4.4, 6.4) 7th | (1.6, 3.4, 5.4) 1st | (4.4, 6.4, 8.1) 7th |
| $A_3$—Starshades in transit | (2.5, 4.4, 6.4) 6th | (3.8, 5.6, 7.3) 4th | (2.9, 4.8, 6.8) 5th | (2.3, 4.2, 6.2) 3rd | (4.2, 6.2, 7.9) 8th |
| $A_4$—Clarke exobelt in transit | (2.0, 3.8, 5.8) 7th | (3.6, 5.4, 7.2) 3rd | (3.1, 5.0, 7.0) 3rd | (1.8, 3.6, 5.6) 2nd | (4.6, 6.6, 8.3) 6th |
| $A_5$.-Laser pulses | (4.8, 6.8, 8.4) 2nd | (1.7, 3.4, 5.4) 1st | (2.8, 4.8, 6.8) 6th | (4.5, 6.4, 8.2) 8th | (6.4, 8.4, 9.6) 1st |
| $A_6$—Heat from megastructures | (3.7, 5.6, 7.5) 3rd | (3.6, 5.6, 7.5) 4th | (3.0, 5.0, 7.0) 4th | (3.9, 5.8, 7.6) 6th | (5.4, 7.4, 9.0) 3rd |
| $A_7$—Radio signals | (5.2, 7.2, 8.7) 1st | (1.8, 3.6, 5.6) 2nd | (2.5, 4.4, 6.4) 8th | (4.2, 6.2, 7.9) 7th | (6.4, 8.4, 9.6) 1st |
| $A_8$—Artifacts orbiting Earth, Moon, or the Sun | (3.2, 5.0, 6.9) 4th | (4.4, 6.4, 8.2) 7th | (4.6, 6.6, 8.4) 1st | (2.9, 4.8, 6.7) 4th | (5.4, 7.4, 8.9) 4th |
| | $C_6$ | $C_7$ | $C_8$ | $C_9$ | $C_{10}$ |
| $A_1$—Industrial gases on atmospheric spectra | (3.8, 5.8, 7.6) 4th | (3.4, 5.2, 7.0) 4th | (3.8, 5.8, 7.8) 2nd | (2.0, 3.8, 5.7) 7th | (3.2, 5.2, 7.2) 4th |
| $A_2$—Dark side illumination | (4.4, 6.4, 8.1) 8th | (3.6, 5.4, 7.2) 5th | (2.8, 4.8, 6.8) 3rd | (2.3, 4.2, 6.2) 5th | (3.2, 5.2, 7.0) 6th |
| $A_3$—Starshades in transit | (4.0, 6.0, 7.9) 7th | (4.0, 6.0, 7.8) 7th | (1.9, 3.6, 5.6) 8th | (2.3, 4.2, 6.1) 6th | (3.4, 5.4, 7.3) 3rd |
| $A_4$—Clarke exobelt in transit | (3.9, 5.8, 7.6) 5th | (3.7, 5.6, 7.4) 6th | (2.6, 4.6, 6.6) 4th | (2.5, 4.4, 6.3) 4th | (3.6, 5.6, 7.4) 2nd |
| $A_5$—Laser pulses | (0.6, 2.0, 4.0) 1st | (2.5, 4.4, 6.4) 2nd | (2.4, 4.4, 6.4) 6th | (5.0, 7.0, 8.7) 3rd | (2.9, 4.8, 6.8) 7th |
| $A_6$—Heat from megastructures | (4.0, 6.0, 7.8) 6th | (2.7, 4.4, 6.2) 2nd | (4.2, 6.2, 7.9) 1st | (1.9, 3.6, 5.6) 8th | (3.2, 5.2, 7.1) 5th |
| $A_7$—Radio signals | (1.1, 2.6, 4.6) 2nd | (2.5, 4.2, 6.2) 1st | (2.1, 4.0, 6.0) 7th | (5.6, 7.6, 9.2) 1st | (2.6, 4.4, 6.4) 8th |
| $A_8$—Artifacts orbiting Earth, Moon, or the Sun | (1.6, 3.2, 5.1) 3rd | (4.6, 6.4, 8.0) 8th | (2.5, 4.4, 6.3) 5th | (5.7, 7.6, 8.9) 2nd | (3.9, 5.8, 7.8) 1st |

## 4. Results and Discussion

Taking into consideration the vector of weights provided by the advisory group, and implementing each of the stages of the fuzzy version of the TOPSIS methodology to such decision matrices (Tables 9 and 10), the prioritization of alternatives for both proposed study cases (categories of TSs and TSs individually) can be generated. Table 11 and Figure 4 show the ranking of the three categories of TSs analyzed, while Table 12 and Figure 5 provide the ranking of the TSs individually considered.

The alternatives that are closest to unity correspond to those that should be prioritized. Therefore, from the perspective of the prioritization of categories of TSs, the group composed of Radio and optical communications (criterion $AG_1$) should receive special attention. However, this preference is not so evident when analyzing the results provided by the remaining categories of TSs ($AG_2$—Waste heat and $AG_3$—Solar System artifacts).

Analyzing the results of the TSs individually, it is worth highlighting the consistency with the ranking of their categories, since the two main TSs according to the advisory group are the alternatives $A_7$—Radio Signals and $A_5$—Laser pulses. It is interesting to highlight the position of alternative $A_3$—Artifacts orbiting Earth, Moon, or the Sun, which we discuss in more detail below. The Clarke exobelt in transit (alternative $A_4$), which occupies the fourth position in the ranking, also demonstrates how the search for communications satellites in geostationary orbit is an interesting option to take into consideration. Finally, it is observed that there is a group composed of four alternatives ($A_1$—Industrial gases on atmospheric spectra, $A_6$—Heat from megastructures, $A_3$—Starshades in transit, and $A_2$—Dark side illumination) whose values in the closeness coefficient of the TOPSIS algorithm do not present notable differences.

Regarding the newly introduced criterion, Scale Sensitivity, experts have ranked it as one of the less critical factors, placing it slightly ahead of the Ancillary Benefits and Cost criteria, but still behind the Merit Axis Extrapolation. Notably, they considered Scale Sensitivity to be most significant in the context of $A_8$—Artifacts orbiting Earth, Moon, or the Sun, followed closely by $A_4$—Clarke exobelt in transit. Conversely, they deemed the technosignatures least influenced by the scale factor to be $A_7$—Radio signal and $A_5$—Laser pulses.

As mentioned, the third-ranked alternative is the existence of artifacts orbiting Earth, the Moon, or the Sun. This proposition raises critical questions regarding the adequacy of the defined merit axes in determining the most effective strategy for SETI. The current criteria fail to directly assess the plausibility of a proposed TS. In essence, these criteria do not sufficiently evaluate the likelihood of the TS's existence. An illustrative extreme example of this limitation is the hypothetical detection of an alien infiltrator within human civilization using extraterrestrial technology. While this TS might have the best scores across all existing merit axes, it is highly questionable due to the extremely low probability of such an occurrence. This scenario underscores a significant gap in the current evaluation framework: the lack of a criterion that directly addresses the realism of a TS's existence.

To rectify this deficiency, we propose an augmentation of the Axes of Merit with an additional criterion focused on the TS's plausibility. This new criterion would serve to evaluate the realistic potential of a TS, ensuring a more comprehensive and pragmatic approach to SETI strategies. By incorporating this dimension, the assessment framework can better balance theoretical optimality with practical feasibility, guiding more realistic and effective SETI. Acknowledging the inherent subjectivity in gauging the plausibility of a TS due to our limited knowledge, it is nevertheless possible to establish certain boundaries for assessment based on our accumulated experience and current understanding of the cosmos (number of objects, distances, times, energy scales, etc.).

**Table 11.** Ranking of the three categories of TSs by crisp (real) numbers.

| Alternatives | Fuzzy Numbers | Defuzzification Process | Ranking |
|---|---|---|---|
| $AG_1$—Radio and optical communications | (0.156, 0.645, 2.660) | 0.900 | 1st |
| $AG_2$—Waste heat | (0.097, 0.430, 1.921) | 0.623 | 3rd |
| $AG_3$—Solar System artefacts | (0.110, 0.462, 1.984) | 0.657 | 2nd |

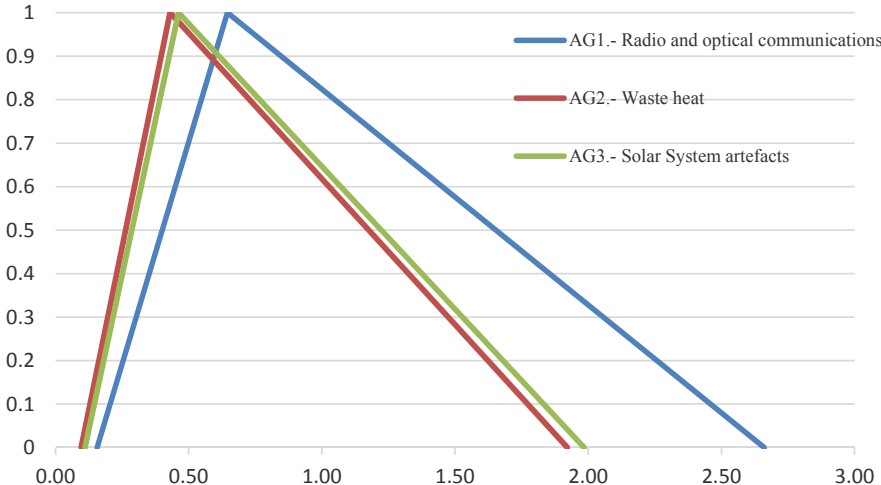

**Figure 4.** Ranking of the three categories of TSs by triangular fuzzy numbers.

**Table 12.** Ranking of TSs by crisp (real) numbers.

| Alternatives | Fuzzy Numbers | Defuzz. Process | Ranking |
|---|---|---|---|
| $A_1$—Industrial gases on atmospheric spectra | (0.093, 0.428, 2.003) | 0.635 | 5th |
| $A_2$—Dark side illumination | (0.093, 0.408, 1.917) | 0.607 | 8th |
| $A_3$—Starshades in transit | (0.096, 0.423, 1.928) | 0.619 | 7th |
| $A_4$—Clarke exobelt in transit | (0.103, 0.454, 2.100) | 0.670 | 4th |
| $A_5$—Laser pulses | (0.123, 0.569, 2.503) | 0.817 | 2nd |
| $A_6$—Heat from megastructures | (0.088, 0.424, 2.025) | 0.635 | 6th |
| $A_7$—Radio signals | (0.134, 0.597, 2.600) | 0.854 | 1st |
| $A_8$—Artifacts orbiting Earth, Moon, or the Sun | (0.131, 0.560, 2.468) | 0.806 | 3rd |

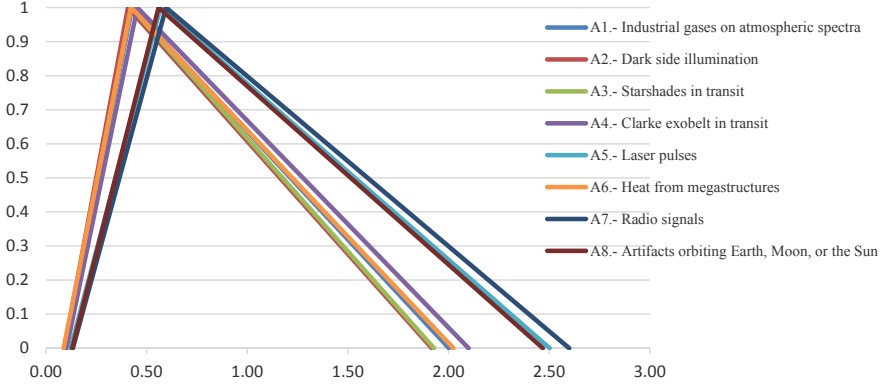

**Figure 5.** Ranking of TSs by triangular fuzzy numbers with defuzzification process values.

## 5. Sensitivity Analysis

The validity of the results achieved in this case study is analyzed and verified through a sensitivity analysis. For this, we analyze the influence of the advisory group which has intervened both in obtaining the weight of the criteria and in the qualitative valuation process of the alternatives. To accomplish this, all alternatives (categories of TSs and individual TSs) are evaluated again through the fuzzy TOPSIS algorithm considering that

all criteria have equal importance. In this way, it is possible to analyze significant variations in the rankings. Tables 13 and 14 and Figures 6 and 7 show the comparison among rankings of alternatives.

**Table 13.** Comparison of the categories of TSs according to the fuzzy TOPSIS method with defuzzification process values.

| Alternatives | Weights by Experts Group | | Criteria with Same Weights | |
| --- | --- | --- | --- | --- |
| | Defuzz. Process | Ranking | Defuzz. Process | Ranking |
| $AG_1$—Radio and optical communications | 0.8995 | 1st | 0.7429 | 1st |
| $AG_2$—Waste heat | 0.6229 | 3rd | 0.5300 | 2nd |
| $AG_3$—Solar System artefacts | 0.6573 | 2nd | 0.4152 | 3rd |

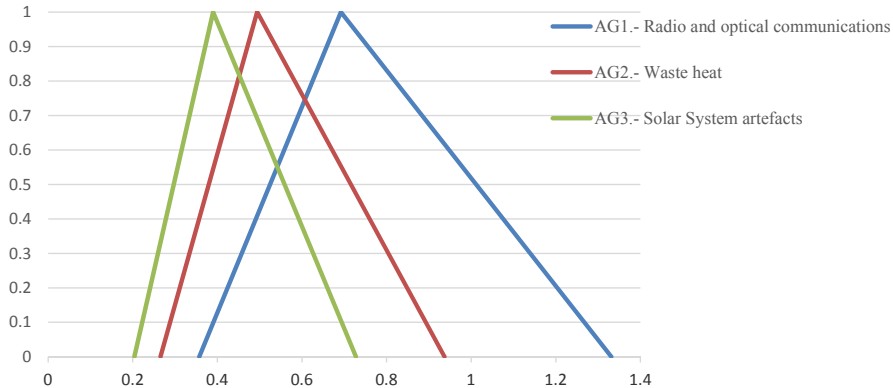

**Figure 6.** Ranking of the categories of TSs by triangular fuzzy numbers considering all the criteria with the same weight.

**Table 14.** Comparison of the TSs according to the fuzzy TOPSIS method with defuzzification process values.

| Alternatives | Weights by Experts Group | | Criteria with Same Weights | |
| --- | --- | --- | --- | --- |
| | Defuzz. Process | Ranking | Defuzz. Process | Ranking |
| $A_1$—Industrial gases on atmospheric spectra | 0.6347 | 5th | 0.4396 | 5th |
| $A_2$—Dark side illumination | 0.6068 | 8th | 0.3339 | 7th |
| $A_3$—Starshades in transit | 0.6192 | 7th | 0.3268 | 8th |
| $A_4$—Clarke exobelt in transit | 0.6698 | 4th | 0.3923 | 6th |
| $A_5$—Laser pulses | 0.8169 | 2nd | 0.7012 | 1st |
| $A_6$—Heat from megastructures | 0.6347 | 6th | 0.4478 | 4th |
| $A_7$—Radio signals | 0.8535 | 1st | 0.6825 | 2nd |
| $A_8$—Artifacts orbiting Earth, Moon, or the Sun | 0.8064 | 3rd | 0.6039 | 3rd |

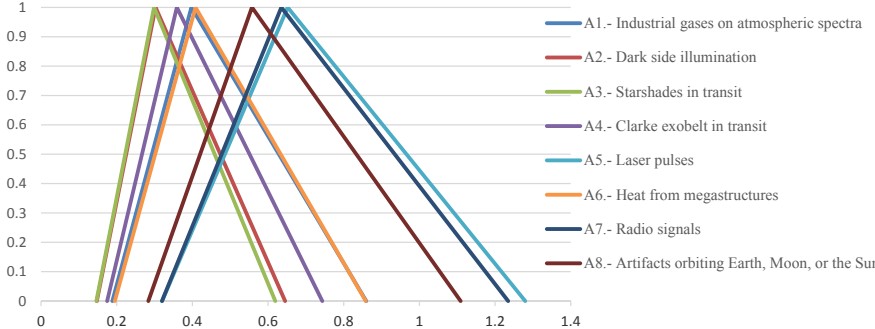

**Figure 7.** Ranking of TSs by triangular fuzzy numbers considering all the criteria with the same weight.

The comparison between the two prioritization rankings of the categories of alternatives shows that the category positioned in first position ($AG_1$—Radio and optical

communications) does not change (see Table 13). This fact confirms the robustness of the evaluations provided by the group of experts through linguistic labels. However, there is an exchange of positions between the other two categories of alternatives ($AG_2$—Waste heat and $AG_3$—Solar System artifacts). This change is not surprising, as the difference in the closeness coefficients of the TOPSIS algorithm for these alternatives is very small when considering the weights provided by the advisory group. Therefore, variations in the weights of the criteria can generate the observed exchange of positions.

In the problem of prioritizing individual TSs, the comparison yields similar results (see Table 14). The robustness of the qualitative evaluations of the criteria for each alternative provided by the group of experts is again corroborated. The first two alternatives ($A_7$—Radio signals and $A_5$—Laser pulses) continue to stand out above the rest. Although there has been an exchange of positions between both, this change can again be justified by the small difference in their closeness coefficients. It is worth highlighting the position of the alternative $A_8$—Artifacts orbiting Earth, Moon, or the Sun, which not only occupies the third position in both rankings but also is reinforced with the values of its corresponding proximity coefficients according to TOPSIS. As a result of this, such alternatives should be the subject of further analysis. It is also worth mentioning, as an alternative, that $A_4$—Clarke exobelt in transit is an interesting option according to the group of experts. This alternative falls to the sixth position if the weights provided by the advisory group (via fuzzy AHP methodology) are not considered.

## 6. Conclusions

As demonstrated, the application of fuzzy MCDM methodologies, specifically the fuzzy TOPSIS method, provides a structured and robust framework for prioritizing TS in SETI efforts. The findings from our analysis, integrating expert opinions and weighted criteria, offer a powerful approach to the SETI field.

The prioritization of TS categories and individual TSs illustrates a nuanced understanding of the relative importance of various TSs. Notably, the prioritization favors radio and optical communications, highlighting their significance in the SETI domain. Nevertheless, the close rankings of other TS categories suggest that a diversified approach may be beneficial. On the other hand, dark side illumination and starshades in transit are the lowest rated by experts.

The sensitivity analysis, considering equal weight for all criteria, further validates the robustness of our approach. While slight variations in rankings were observed, the overall consistency in prioritization underlines the reliability of the expert evaluations.

The results of this study, particularly the emphasis placed on artifacts orbiting Earth, the Moon, or the Sun, and the Clarke exobelt in transit, underscore in our opinion the necessity to broaden the criteria for prioritizing the search for extraterrestrial life. This, coupled with the introduction and subsequent ranking of the Scale Sensitivity criterion, highlights the evolving landscape of SETI research. It suggests that some further revision to the Axes of Merit may be beneficial in our ongoing quest to explore extraterrestrial existence.

As we advance in our quest to answer the profound question of our solitude in the cosmos, this study exemplifies the need for continuous refinement and expansion of our methodologies. The proposed augmentation of the Axes of Merit to include criteria that directly address the plausibility of technosignatures is a step in this direction. It balances theoretical rigor with practical feasibility, enhancing the capabilities of the SETI endeavor.

In light of these findings, future research should focus on refining the criteria for technosignature identification, exploring new methodologies for prioritization, and expanding the scope of SETI to include a broader range of technosignatures. As we advance in our quest to answer the profound question of our solitude in the cosmos, new approaches will have to uncover the mysteries of extraterrestrial life in the vast expanse of the universe.

**Author Contributions:** Conceptualization, E.P.-A.; methodology, J.M.S.-L.; writing J.M.S.-L. and E.P.-A.; supervision, review, and editing H.S.-N. All authors have read and agreed to the published version of the manuscript.

**Funding:** The APC was funded by the project Fundación Séneca (grant number: 22069/PI/22), Spain. J.M.S.-L. and E.P.-A. have carried out this work in the framework of the project Fundación Séneca (grant number: 22069/PI/22), Spain. J.M.S.-L. thanks funding from the Ministerio de Ciencia e Innovación (Spain) (grant numbers: PID2020-112754GB-I00 and PID2021-128062NB-I00). E.P.-A. thanks funding from the European Research Council (ERC) under the European Union's Horizon 2020 Research and Innovation Programme (grant agreement No. 865657) for the project "Quantum Chemistry on Interstellar Grains" (QUANTUMGRAIN). E.P.-A. also recognizes partial support from the program Unidad de Excelencia María de Maeztu CEX2020-001058-M. H.S.-N. acknowledges support from the Agencia Estatal de Investigación del Ministerio de Ciencia e Innovación (AEI-MCINN) under grant Hydrated Minerals and Organic Compounds in Primitive Asteroids with reference PID2020-120464GB-I100.

**Data Availability Statement:** The data presented in this study are available on request from the corresponding author.

**Conflicts of Interest:** The authors declare no conflicts of interest.

## Abbreviations

The following abbreviations are used in this manuscript:

| | |
|---|---|
| TS | Technosignature |
| SETI | Search for Extraterrestrial Intelligence |
| MCDM | Multi-Criteria Decision Making |
| AHP | Analytic Hierarchy Process |
| TOPSIS | Technique for Order of Preference by Similarity to Ideal Solution |

## Appendix A

This appendix presents a detailed overview of the evaluation criteria weights assigned by the advisory group, as delineated in Table A1. Additionally, it encompasses the individual responses of the experts, segmented into two distinct categories; the collective responses by category are displayed in Table A2, while the individual responses for each technosignature are systematically arranged in Table A3.

**Table A1.** Weights of criteria of the advisory group.

| Criteria | Expert 1 ($E_2$) | Expert 2 ($E_2$) | Expert 3 ($E_3$) | Expert 4 ($E_4$) | Expert 5 ($E_5$) |
|---|---|---|---|---|---|
| $C_1$ | (0.193, 0.337, 0.547) | (0.258, 0.395, 0.583) | (0.040, 0.063, 0.097) | (0.136, 0.188, 0.248) | (0.042, 0.062 0.090) |
| $C_2$ | (0.052, 0.112, 0.255) | (0.033, 0.056, 0.094) | (0.040, 0.063, 0.097) | (0.036, 0.063, 0.116) | (0.036, 0.048 0.069) |
| $C_3$ | (0.025, 0.048, 0.089) | (0.033, 0.056, 0.094) | (0.040, 0.063, 0.097) | (0.036, 0.063, 0.116) | (0.036, 0.048 0.069) |
| $C_4$ | (0.052, 0.112, 0.255) | (0.069, 0.132, 0.272) | (0.314, 0.438, 0.600) | (0.136, 0.188, 0.248) | (0.056, 0.087 0.134) |
| $C_5$ | (0.034, 0.067, 0.131) | (0.033, 0.056, 0.094) | (0.040, 0.063, 0.097) | (0.136, 0.188, 0.248) | (0.324, 0.435 0.557) |
| $C_6$ | (0.052, 0.112, 0.255) | (0.033, 0.056, 0.094) | (0.040, 0.063, 0.097) | (0.015, 0.021, 0.031) | (0.056, 0.087 0.134) |
| $C_7$ | (0.022, 0.037, 0.068) | (0.033, 0.056, 0.094) | (0.040, 0.063, 0.097) | (0.015, 0.021, 0.031) | (0.036, 0.048 0.069) |
| $C_8$ | (0.034, 0.067, 0.131) | (0.045, 0.079, 0.140) | (0.040, 0.063, 0.097) | (0.136, 0.188, 0.248) | (0.056, 0.087 0.134) |
| $C_9$ | (0.022, 0.037, 0.068) | (0.033, 0.056, 0.094) | (0.040, 0.063, 0.097) | (0.036, 0.063, 0.116) | (0.036, 0.048 0.069) |
| $C_{10}$ | (0.034, 0.067, 0.131) | (0.033, 0.056, 0.094) | (0.040, 0.063, 0.097) | (0.015, 0.021, 0.031) | (0.036, 0.048 0.069) |
| **Criteria** | **Expert 6 ($E_6$)** | **Expert 7 ($E_7$)** | **Expert 8 ($E_8$)** | **Expert 9 ($E_9$)** | **Expert 10 ($E_{10}$)** |
| $C_1$ | (0.248, 0.386, 0.579) | (0.040, 0.063, 0.097) | (0.178, 0.209, 0.242) | (0.070, 0.131, 0.258) | (0.037, 0.049, 0.071) |
| $C_2$ | (0.032, 0.055, 0.094) | (0.040, 0.063, 0.097) | (0.023, 0.030, 0.039) | (0.029, 0.044, 0.068) | (0.037, 0.049, 0.071) |
| $C_3$ | (0.032, 0.055, 0.094) | (0.040, 0.063, 0.097) | (0.020, 0.023, 0.030) | (0.029, 0.044, 0.068) | (0.043, 0.064, 0.094) |
| $C_4$ | (0.066, 0.129, 0.270) | (0.314, 0.438, 0.600) | (0.178, 0.209, 0.242) | (0.262, 0.394, 0.554) | (0.043, 0.064, 0.094) |
| $C_5$ | (0.043, 0.077, 0.139) | (0.040, 0.063, 0.097) | (0.023, 0.030, 0.039) | (0.034, 0.056, 0.090) | (0.058, 0.089, 0.139) |
| $C_6$ | (0.032, 0.055, 0.094) | (0.040, 0.063, 0.097) | (0.178, 0.209, 0.242) | (0.034, 0.056, 0.090) | (0.043, 0.064, 0.094) |
| $C_7$ | (0.032, 0.055, 0.094) | (0.040, 0.063, 0.097) | (0.023, 0.030, 0.039) | (0.034, 0.056, 0.090) | (0.043, 0.064, 0.094) |
| $C_8$ | (0.032, 0.055, 0.094) | (0.040, 0.063, 0.097) | (0.020, 0.023, 0.030) | (0.070, 0.131, 0.258) | (0.043, 0.064, 0.094) |
| $C_9$ | (0.032, 0.055, 0.094) | (0.040, 0.063, 0.097) | (0.178, 0.209, 0.242) | (0.029, 0.044, 0.068) | (0.332, 0.445, 0.578) |
| $C_{10}$ | (0.043, 0.077, 0.139) | (0.040, 0.063, 0.097) | (0.023, 0.030, 0.039) | (0.029, 0.044, 0.068) | (0.037, 0.049, 0.071) |

**Table A2.** Decision matrix of alternatives and criteria (three TS categories) Note: Criteria with a positive sign (+) correspond to criteria to be maximized, while those with a negative sign (−) must be minimized.

| Experts | Alternatives | Criteria | | | | | | | | | |
|---------|--------------|----------|----------|----------|----------|----------|----------|----------|----------|----------|-----------|
| | | $C_1^+$ | $C_2^-$ | $C_3^+$ | $C_4^-$ | $C_5^+$ | $C_6^-$ | $C_7^-$ | $C_8^+$ | $C_9^+$ | $C_{10}^+$ |
| $E_1$ | | VH | L | H | VH | B | L | VL | M | H | H |
| $E_2$ | | VH | M | H | VL | VB | VL | VL | VH | VH | M |
| $E_3$ | | M | M | L | L | B | VL | H | L | VH | VH |
| $E_4$ | | VH | VL | VL | M | VB | VL | L | VL | VH | L |
| $E_5$ | $AG_1$—Radio and Optical | VH | M | M | VH | M | H | VH | VH | H | M |
| $E_6$ | | VH | L | H | M | M | M | M | M | H | M |
| $E_7$ | | M | M | L | M | M | M | M | M | L | M |
| $E_8$ | | VH | VL | VH | VH | VH | VL | VL | VH | VH | VL |
| $E_9$ | | M | L | M | M | VB | H | L | L | M | H |
| $E_{10}$ | | VH | VL | H | M | M | M | VL | M | H | VH |
| $E_1$ | | H | H | H | H | VH | M | VH | H | H | M |
| $E_2$ | | VH | M | M | L | H | VH | H | H | L | VH |
| $E_3$ | | H | H | H | H | H | VH | VH | VH | VL | H |
| $E_4$ | | M | L | M | M | VH | L | VL | VH | L | H |
| $E_5$ | $AG_2$—Waste Heat | VH | M | L | M | VH | L | L | H | M | H |
| $E_6$ | | M | M | M | H | H | H | L | M | M | H |
| $E_7$ | | H | M | H | VH | VH | H | VL | VH | VH | M |
| $E_8$ | | L | H | VH | L | M | VH | M | M | VL | VH |
| $E_9$ | | H | M | M | H | H | H | H | VH | L | VH |
| $E_{10}$ | | H | L | H | M | VH | M | H | M | H | VH |
| $E_1$ | | M | M | H | VL | VH | VL | H | L | VH | H |
| $E_2$ | | M | VH | L | H | VH | VL | VH | M | VH | M |
| $E_3$ | | H | H | H | M | H | L | H | L | VH | VH |
| $E_4$ | | H | H | H | L | H | H | M | M | H | M |
| $E_5$ | $AG_3$—Solar System Artifacts | M | M | VH | L | M | M | H | L | VH | M |
| $E_6$ | | M | H | M | VH | B | VH | VH | L | VH | VH |
| $E_7$ | | M | M | H | M | M | H | VH | M | H | M |
| $E_8$ | | VL | VH | VH | VL | M | M | VH | VL | L | VH |
| $E_9$ | | VH | H | VH | H | M | VL | H | VL | VH | VL |
| $E_{10}$ | | M | M | VH | L | VH | VL | M | VL | VH | L |

**Table A3.** Decision matrix of alternatives and criteria (TSs individually). Note: Criteria with a positive sign (+) correspond to criteria to be maximized, while those with a negative sign (−) must be minimized.

| Experts | Alternatives | Criteria | | | | | | | | | |
|---------|--------------|----------|----------|----------|----------|----------|----------|----------|----------|----------|-----------|
| | | $C_1^+$ | $C_2^-$ | $C_3^+$ | $C_4^-$ | $C_5^+$ | $C_6^-$ | $C_7^-$ | $C_8^+$ | $C_9^+$ | $C_{10}^+$ |
| $E_1$ | | VL | H | H | L | M | VH | VL | H | L | L |
| $E_2$ | | H | M | M | M | VH | M | M | H | M | H |
| $E_3$ | | L | VH | VH | H | M | M | L | M | L | H |
| $E_4$ | | L | VL | H | M | VB | H | H | H | VL | M |
| $E_5$ | $A_1$—Industrial gases | VH | H | M | M | H | L | H | M | L | L |
| $E_6$ | on atmospheric spectra | H | H | M | H | H | L | M | M | M | M |
| $E_7$ | | M | M | M | M | M | M | M | M | M | M |
| $E_8$ | | VL | VH | M | VL | M | VH | VH | M | VL | M |
| $E_9$ | | H | M | VH | H | H | H | VH | H | VH | M |
| $E_{10}$ | | H | M | H | M | VH | M | VL | M | L | H |

**Table A3.** *Cont.*

| Experts | Alternatives | Criteria | | | | | | | | | |
|---|---|---|---|---|---|---|---|---|---|---|---|
| | | $C_1^+$ | $C_2^-$ | $C_3^+$ | $C_4^-$ | $C_5^+$ | $C_6^-$ | $C_7^-$ | $C_8^+$ | $C_9^+$ | $C_{10}^+$ |
| $E_1$ | | VL | H | H | VL | M | VH | VL | H | L | L |
| $E_2$ | | L | M | VL | L | VH | H | H | M | H | VH |
| $E_3$ | | VL | VH | H | M | M | L | M | M | L | H |
| $E_4$ | | L | VL | M | L | VB | M | H | L | L | M |
| $E_5$ | $A_2$—Dark side illumination | H | M | L | M | M | L | H | M | L | L |
| $E_6$ | | L | VH | H | L | M | H | L | L | H | L |
| $E_7$ | | M | M | M | M | M | M | M | M | M | M |
| $E_8$ | | VL | VH | M | VL | M | VH | VH | M | VL | M |
| $E_9$ | | VL | VH | VL | M | H | H | VH | M | H | L |
| $E_{10}$ | | L | H | L | L | VH | VH | VL | M | L | VH |
| $E_1$ | | M | M | H | L | M | H | L | M | L | M |
| $E_2$ | | H | M | L | H | VH | H | M | M | VH | M |
| $E_3$ | | M | M | H | H | M | L | H | VL | L | H |
| $E_4$ | | L | VL | M | L | B | M | M | M | L | M |
| $E_5$ | $A_3$—Starshades in transit | M | H | VL | L | M | M | M | VL | L | L |
| $E_6$ | | L | VH | H | L | M | H | H | L | H | L |
| $E_7$ | | M | M | M | M | M | M | M | M | M | M |
| $E_8$ | | VL | VH | M | VL | M | VH | VH | M | VL | M |
| $E_9$ | | M | VH | M | M | M | H | M | M | M | H |
| $E_{10}$ | | M | VL | L | M | VH | M | VH | VL | L | VH |
| $E_1$ | | M | M | H | M | M | H | L | H | L | L |
| $E_2$ | | VL | M | M | VL | H | H | VH | L | VH | VH |
| $E_3$ | | L | H | H | M | M | M | H | L | L | H |
| $E_4$ | | L | VL | M | L | B | M | M | M | L | M |
| $E_5$ | $A_4$—Clarke exobelt in transit | H | M | VL | L | M | L | M | L | L | L |
| $E_6$ | | L | VH | H | L | H | H | M | L | H | L |
| $E_7$ | | M | M | M | M | M | M | M | M | M | M |
| $E_8$ | | VL | VH | M | VL | M | VH | VH | M | VL | M |
| $E_9$ | | M | H | M | M | VH | VH | H | H | H | VH |
| $E_{10}$ | | M | VL | L | M | VH | VL | VL | M | L | H |
| $E_1$ | | VH | VL | H | VH | B | L | H | M | H | H |
| $E_2$ | | M | M | M | H | B | VL | L | L | H | L |
| $E_3$ | | M | L | L | M | B | VL | H | L | VH | H |
| $E_4$ | | H | L | M | VL | VB | VL | L | H | VH | M |
| $E_5$ | $A_5$—Laser pulses | VH | L | M | H | VB | VL | L | L | H | M |
| $E_6$ | | M | M | M | H | H | VL | M | H | H | H |
| $E_7$ | | M | M | M | M | M | M | M | M | M | M |
| $E_8$ | | VH | VL | M | VH | VH | VL | VL | M | VH | VL |
| $E_9$ | | M | H | M | H | VB | L | H | L | H | M |
| $E_{10}$ | | VH | VL | L | H | VB | L | L | L | L | L |
| $E_1$ | | H | H | H | H | H | M | L | H | VL | M |
| $E_2$ | | H | M | L | VH | H | H | VL | H | L | L |
| $E_3$ | | H | H | H | H | H | VH | VH | VH | VL | H |
| $E_4$ | | H | L | M | VH | VH | H | VL | VH | L | H |
| $E_5$ | $A_6$—Heat from | L | M | L | L | H | H | VL | L | M | L |
| $E_6$ | megastructures | M | M | M | M | VH | L | L | M | H | M |
| $E_7$ | | M | M | M | M | M | M | M | M | M | M |
| $E_8$ | | VL | VH | M | VL | M | VH | VH | M | VL | M |
| $E_9$ | | VH | M | H | H | VH | L | M | VH | H | L |
| $E_{10}$ | | M | M | L | M | VH | M | H | L | L | VH |

**Table A3.** *Cont.*

| Experts | Alternatives | Criteria | | | | | | | | | |
|---------|--------------|----------|----------|----------|----------|----------|----------|----------|----------|----------|----------|
| | | $C_1^+$ | $C_2^-$ | $C_3^+$ | $C_4^-$ | $C_5^+$ | $C_6^-$ | $C_7^-$ | $C_8^+$ | $C_9^+$ | $C_{10}^+$ |
| $E_1$ | | VH | L | H | VH | B | L | H | M | H | H |
| $E_2$ | | VH | L | H | VH | VB | VL | VL | M | VH | VL |
| $E_3$ | | VH | L | L | M | B | VL | M | L | VH | H |
| $E_4$ | | H | M | L | L | B | VL | H | L | VH | M |
| $E_5$ | $A_7$—Radio signals | M | H | VL | M | VB | M | M | M | H | L |
| $E_6$ | | M | M | M | H | H | L | L | M | H | H |
| $E_7$ | | M | M | M | M | M | M | M | M | M | M |
| $E_8$ | | VH | VL | M | VH | VH | VL | VL | M | VH | VL |
| $E_9$ | | M | L | M | L | VB | M | H | VL | H | M |
| $E_{10}$ | | VH | VL | L | H | B | VL | VL | L | H | L |
| $E_1$ | | M | M | H | M | VH | VL | VH | VL | VH | M |
| $E_2$ | | H | H | H | M | VH | L | H | VH | VH | H |
| $E_3$ | | H | M | H | M | H | L | H | L | VH | H |
| $E_4$ | | M | M | H | H | VH | VL | H | M | VH | H |
| $E_5$ | $A_8$—Artifacts orbiting Earth, | VL | H | VH | L | M | M | VL | L | H | H |
| $E_6$ | Moon, or the Sun | H | H | H | M | H | L | VH | L | VH | H |
| $E_7$ | | M | M | M | M | M | M | M | M | M | M |
| $E_8$ | | VL | VH | M | VL | M | VH | VH | M | VL | M |
| $E_9$ | | VH | VH | VH | VH | VH | VL | VH | H | VH | VL |
| $E_{10}$ | | L | M | L | L | VH | VL | VL | L | VH | H |

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
