# Peer review of "Deciding Technosignature Search Strategies: Multi-Criteria Fuzzy Logic to Find Extraterrestrial Intelligence"

_aerospace, doi:10.3390/aerospace11010088_

Round 1
Reviewer 1 Report
Comments and Suggestions for Authors
see the attachment.

The language should be further improved with the help of a native to avoid any typos and grammar mistakes.
Author Response
Dear Reviewers,
We appreciate the time and effort dedicated for reviewing our manuscript. Your constructive feedback has been instrumental in enhancing the quality of our work. We have addressed each comment systematically, as detailed below. In the revised manuscript, changes are highlighted in red for ease of identification. Our responses are presented in Times New Roman font.
Reviewer 1
The paper proposed to use fuzzy AHP-TOPSIS for deciding technosignature search strategies. Overall, it is an interesting application study. I suggest the authors further revise the paper based on the following comments:
1) Introduction section, I suggest the authors clarify the reason why AHP and TOPSIS methods are selected instead of other MCDM methods.
- We have added to the text this explanation to to clarify the reasons why we chose AHP and TOPSIS:
“In this study, we select a combination of fuzzy versions of MCDM methodologies (AHP and TOPSIS) due to their complementary strengths. AHP is utilized for its ability to decompose any decision-making problem into a structured hierarchy encompassing objectives, criteria, and alternatives, thereby simplifying the complexity of the problem. Its use of pairwise comparisons for criteria also adds an intuitive element to the evaluation process. Concurrently, the TOPSIS method is employed for its logical framework, which facilitates the representation of both the criteria involved in the decision-making process and their respective significance coefficients through straightforward mathematical procedures. The suitability and justification of the proposed methodology is detailed in the respective sections of this article.”
2) Section 2.2 is called AHP, but the authors introduce fuzzy AHP, which is inconsistent.
- We have changed the named of the section to “The fuzzy analytic hierarchy process (AHP)”.
3) You use C_i and C_jtodenote criteria and also denote the judgment matrix as C, which may cause confusion.
- We have also change the terminology of the matrix from C to H.
4) For Eq. 2, the w_i is a triangular fuzzy number, so how it can be integrated with fuzzy TOPSIS in the following section?
- We haved added the following explanation to Eq. 2:
“Furthermore, in determining the weights of the criteria whose nature is qualitative, the normalized geometric mean based on triangular fuzzy numbers can be applied.
2
This expression, represented by triangular fuzzy numbers, allows us to approximate the calculation of the eigenvector of the matrix H, directly obtaining the weight vector.”
5) For Eq. 3, x_ij is a triangular fuzzy number, so how the values are normalized. As with other steps, the authors should introduce a fuzzy version of TOPSIS instead of a classical TOPSIS. What is the meaning of w_ij in Eq. 4? I guess it should be w_j.
- We haved added this clarification:
“For simplicity of representation, the stages of the TOPSIS algorithm are shown with real values or crisp numbers. The only difference concerning the fuzzy version lies in applying the arithmetic operations of triangular fuzzy numbers, detailed in García-Cascales et al. 2012 and Triantaphyllou 2000.” - We have corrected w_j.
6) The first line on page 8, 10 should be C_10.
7) The authors should provide some comparative studies to justify the proposed method.
- We now cite relevante scientific literature that justify the implementation of the proposed method:
“Moreover, the application of the fuzzy versions, based on triangular fuzzy numbers, of the MCDM methodologies proposed in this study have already been used to solve study cases as diverse as the selection of onshore wind farms (Sánchez-Lozano et al. 2016), evaluation of near-Earth asteroid deflection techniques (Sánchez-Lozano et al. 2020) or the assessment of international military high-performance aircraft (Sánchez-Lozano et al. 2022).”
8) As the paper is related to decision making with linguistic information and AHP, some related studies can be cited and commented in your paper, for instance, Consensus reaching with consistency control in group decision-making with incomplete hesitant fuzzy linguistic preference relations; Consensus reaching for MAGDM with multi-granular hesitant fuzzy linguistic term sets: a minimum adjustment-based approach; Consistency improvement for fuzzy preference relations with self-confidence: An application in two-sided matching decision making.
- We have added these references, which we believe they are in line with and complement the work.
“In fact, the preferences of decision-makers have been frequently analyzed in the literature, current examples are the models for managing incomplete information and consensus with hesitant fuzzy linguistic preference relations in group decision-making problems (Zhuolin et al. 2022), new approaches based on multi-granular hesitant fuzzy linguistic term sets (Wenyu et al. 2021), or even extended logarithmic least squares methods to derive a priority weight vector from a fuzzy preference relation with self-confidence (Zhen et al. 2021)”
9) The language should be further improved with the help of a native to avoid any typos and grammar mistakes.
- We have revised and improved grammar and spelling.

Reviewer 2 Report
Comments and Suggestions for Authors
Dear authors,
Thank you for sharing your paper "Deciding technosignature search strategies: Multi-criteria fuzzy logic to find extraterrestrial intelligence". I think your approach has good potential, but as it stands, the work is hard to follow. Let me start with some general comments that would improve the quality of the work
1. Explain why you chose 10 experts?
2. Explain why you decided on triangular membership functions?
3. Lines 142-145: Explain why "the normalized geometric mean based on triangular fuzzy numbers must be applied"?
4. Line 198-199. Why did you decide to defuzzify using expression (10)?
Author Response
Dear Reviewers,
We appreciate the time and effort dedicated for reviewing our manuscript. Your constructive feedback has been instrumental in enhancing the quality of our work. We have addressed each comment systematically, as detailed below. In the revised manuscript, changes are highlighted in red for ease of identification. Our responses are presented in Times New Roman font.
Reviewer 2
Dear authors,
Thank you for sharing your paper "Deciding technosignature search strategies: Multi-criteria fuzzy logic to find extraterrestrial intelligence". I think your approach has good potential, but as it stands, the work is hard to follow. Let me start with some general comments that would improve the quality of the work
- Explain why you chose 10 experts?
- These were the experts who decided to participate; it is not a predefined value, but it is in accordance with the number of experts usually consulted in this type of study. We have added this paragraph to the manuscript:
“Although ten decision-makers who decided to participate is an adequate number to undertake studies of this nature, there is no specific number of experts that must be gathered. In this type of process, where the decision problem is so specific, the expertise and knowledge of each of the experts involved is much more important than having the participation of a large number of experts.”
- Explain why you decided on triangular membership functions?
- We have added the following paragraphs with references to justify our methodology:
“In fact, the preferences of decision.makers have been frequently analyzed in the literature, current examples are the models for managing incomplete information and consensus with hesitant fuzzy linguistic preference relations in group decision-making problems (Zhuolin et al. 2022), new approaches based on multi-granular hesitant fuzzy linguistic term sets (Wenyu et al. 2021), or even extended logarithmic least squares methods to derive a priority weight vector from a fuzzy preference relation with self-confidence (Zhen et al. 2021). Although these functions, and even new extensions, are being developed today, the advantage of using some over others has not yet been demonstrated, even recent studies have shown that new extensions of fuzzy MCDM versions generate greater dependence on the judgments provided by decision-makers (Sánchez-Lozano & Fernandez-Martinez, 2021). Also starting from the premise that it is unnecessary to increase the complexity of the calculation process, fuzzy sets based on triangular membership functions are applied in this study. This type of function not only simplifies the calculation process due to its easy handling but also fits with the way of representing the qualitative criteria. Moreover, the application of the fuzzy versions, based on triangular fuzzy numbers, of the MCDM methodologies proposed in this study has already been used to solve study cases as diverse as the selection of onshore wind farms (Sánchez-Lozano et al. 2016), evaluation of near-Earth asteroid deflection techniques (Sanchez-Lozano et al. 2020) or the assessment of international military high-performance aircraft (Sánchez-Lozano et al. 2022).”
- Lines 142-145: Explain why "the normalized geometric mean based on triangular fuzzy numbers must be applied"?
- We have added the following explanation to Eq. 2:
“Furthermore, in determining the weights of the criteria whose nature is qualitative, the normalized geometric mean based on triangular fuzzy numbers can be applied.
2
This expression, represented by triangular fuzzy numbers, allows us to approximate the calculation of the eigenvector of the matrix H, directly obtaining the weight vector.”
- Line 198-199. Why did you decide to defuzzify using expression (10)?
- We have added cited to justify this approach and added the following text:
“This defuzzification index allows us to rank fuzzy numbers by modality and optimistic of the decision-maker attitude. Its suitability has been demonstrated thanks to its application in previous studies (García-Cascales et al. 2012, Sánchez-Lozano et al. 2016, Sánchez-Lozano et al. 2022). More detailed information about such index can be seen at Cascales et al. 2007.”

Round 2
Reviewer 1 Report
Comments and Suggestions for Authors
The paper can be considered for publication.
Comments on the Quality of English LanguageMinor editing of English language may be required.